# Multifaceted Pathways of Carbon Emission Reduction Policies: A Study Based on the Decoupling Effect and fsQCA Method in the Yangtze River Delta Urban Agglomeration

**Qianqi Xu [1], Peng Xu [2] and Cunkuan Bao [1,\*]**

[1] Department of Environmental Science and Engineering, Fudan University, Shanghai 200438, China
[2] School of Public Administration, Xi'an University of Architecture and Technology, Xi'an 710055, China
\* Correspondence: baock@fudan.edu.cn; Tel.: +86-130-4467-3815

**Abstract:** The urban carbon administration discussion is becoming dominant in the carbon emission reduction landscape. China, the world's paramount carbon emitter, has instituted a series of policies geared towards carbon emissions reduction. The co-dependent nature of these policies becomes paramount in outlining accurate and comprehensive evaluation results. This research scrutinizes the effect of various policy mixtures on curbing urban carbon emissions, utilizing the Tapio decoupling elasticity coefficient and fuzzy set qualitative comparative analysis (fsQCA) techniques. The consistencies surpass the widely accepted theoretical threshold of 0.8, and each configuration can explain 14.3% to 19.8% of the sample. The findings reveal that industrial and energy policies drive the decoupling of urban carbon emissions in three of four configurations. Management-based policies merely play an auxiliary role, while social and economic policies have yet to exhibit significant implications. The outcomes of this research offer a framework for augmenting the carbon emission reduction policy system and fostering the advancement of carbon reduction initiatives.

**Keywords:** sustainable urban development; carbon emissions; fsQCA; Tapio decoupling model

## 1. Introduction

Global climate change is a predominant challenge currently confronting the world. Mitigating greenhouse gas emissions has garnered considerable attention and significance from nations worldwide. As the world's leading emitter of carbon dioxide, China's carbon reduction initiatives will significantly contribute to global climate change mitigation. To fulfill its commitment to peak carbon emissions, the Chinese government has implemented a series of policies. China's carbon governance system continues to advance, transitioning from singular directives towards a coordinated cobweb of varied policies. The effectiveness of these policies is not considered in isolation but rather evaluated collectively. Different policies may reinforce or complement each other, and in some instances, may overlap or even contradict each other [1–3]. Certain policy implementations may also indirectly influence urban carbon emissions, an example being the utilization of materials with lower carbon footprints in urban construction [4–6]. Particularly with respect to environmental matters like carbon reduction, which involves complex factors such as energy and economics, policies must be synergistic to function effectively in a holistic manner [7–9]. Current research on the effects of carbon reduction policy implementation primarily relies on quasi-natural experiments, comparing similar units after policy implementation. These include policies on low-carbon pilots [10–14], carbon trading [15–20], carbon taxes [21–23], and energy conservation [24–26]. However, in-depth studies on the impacts of the combination of these carbon reduction policies are relatively scarce.

Cities are the centers of economic activity and significant contributors to carbon emissions. According to statistics from Cai et al., urban carbon emissions account for approximately 70% of China's total carbon emissions [27]. This paper explores the mechanism

of how the policy mix of urban carbon emission reduction influences the carbon diminution in cities. Although the Yangtze River Delta (YRD) urban agglomeration occupies merely 2.3% of China's total land area, it is home to 225 million individuals and generates approximately a quarter of the nation's GDP. This dynamic region is confronted with intense carbon emission pressures due to its rapid economic growth. Analyzing the carbon emission policy mix of the YRD can facilitate the design of more targeted emission reduction strategies and measures. Furthermore, such analysis can serve as a valuable reference, offering insights to other regions and cities for promoting carbon emission reduction. The location of YRD is shown in Figure 1.

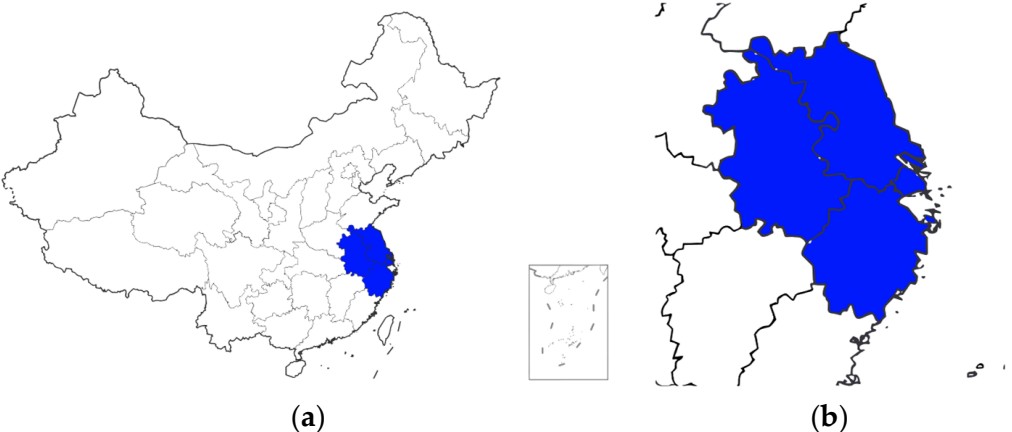

(**a**)                                             (**b**)

**Figure 1.** (**a**) The location of the Yangtze River Delta. (**b**) An enlarged map showing the four provincial administrative regions.

Leveraging Tapio's decoupling theory and fuzzy set qualitative comparative analysis (FsQCA), this paper focuses on cities within the YRD region. The intent is to assess the multifaceted influence of diverse policy frameworks on the decoupling between urban carbon reduction and economic advancement, with the ultimate goal of augmenting carbon reduction policies. The study regards the carbon reduction policies in 41 cities in the YRD region as antecedent conditions, while modifications in urban carbon emission decoupling indices from 2015 to 2020 are treated as outcome variables. The exploration of the correlation between various policy configurations and the enhancement of urban decoupling indices is achieved through causal analysis.

The paper is structured as follows: Section 2 presents a literature review, outlining the study's theoretical basis and research design; Section 3 specifies the data sources and methodology; Section 4 provides the study's results; Section 5 deliberates on the findings; and Section 6 summarizes the main conclusions, followed by corresponding recommendations based on these insights.

## 2. Literature Review

### 2.1. Policy Mix

The effectiveness of policy implementation is not solely attributed to an individual policy, but instead emerges from the collective action of various policies [28]. Interactions between different policies can either amplify—wherein the impact of one policy bolsters the impact of another [1,29]—or they might exhibit complementary effects, collaboratively contributing to the achievement of objectives. Conversely, there could be instances of duplicative or even antagonistic effects among different policies. The latter implies redundant pursuit of the same objectives, resulting in unnecessary resource allocation and effort duplication, or conflicting policies that obstruct each other's implementation [30]. Consequently, one must consider these policy interactions while constructing and implementing policies to ascertain the ideal policy mixture and harness their combined effects optimally. The present research on carbon emission reduction policy effects primarily focuses on

post-implementation scenarios of certain policy types. For instance, studies delve into the impacts of low-carbon pilot policies [10–14], efficiency evaluation of carbon trading policies [15–20,24], the effectiveness and impact mechanisms of energy-saving policies [24–26], and the implementation effects of a carbon tax policy [21–23]. Simultaneously, influences of policies outside the carbon reduction sphere like economic policy [31], energy policy [32], and transportation policy on carbon emission reduction are also noted [5]. Nonetheless, less attention has been given to in-depth analysis on the impacts of carbon emission reduction policy combinations. Therefore, comprehensive research is warranted to comprehend how different carbon emission policy combinations influence the environment and economy.

Policy mix theory effectively addresses the insufficiencies in the coverage of dimensionality within policy research [1,3,28,30]. A policy mix is typically defined as a combination of policies or policy instruments that interact with each other to influence the outcomes of policy implementation [28,33]. The complexity and multitude of public policies, coupled with the problem of fragmented solutions for a single issue across disparate policy documents, present a considerable challenge in public policy. To mitigate this, policy mix theory may be employed to lucidly delineate the policy network structure within governance units. Upon analysis of policy mix theory, it becomes apparent that the urban carbon emission reduction policy mix is a policy system dedicated to carbon emission reduction, encompassing sub-policy systems such as industry, energy, economy, management, society, and technology [2,7,34]. Generally, policies do not function independently, instead establishing direct or indirect linkages with policies from various domains [8].

### 2.2. Classification of Carbon Emission Reduction Policies

Presently, different academicians offer diverse classifications for carbon emission reduction policies. Some bifurcate these into regulatory, economic, and social types [35], drawing from aspects such as governmental enforcement, market regulation, and contributions from assorted social entities. Additional scholars arrange these policies based on the areas they pertain to, such as energy-related, industrial, or varying emission domains [36]. Interestingly, some researchers structure low-carbon policies based on their primary focus. This includes energy policies that aim towards restructuring energy dynamics, technology-oriented policies that stimulate low-carbon technological innovation, industrial policies that foster low-carbon industrial progression and product creation, and consumption policies that advocate for green consumption [37]. With the goal of exploring the cumulative effect of varying policy types on the decoupling impact, this study, referring to the aforementioned classifications, further divides carbon emission reduction policies into six predominant categories: "industrial", "economic", "energy", "management-based", "social", and "technology-oriented" [17,38].

Industrial and energy policies currently form the cornerstone of China's carbon emission management, with the industrial structure acting as a primary influencer of urban carbon emissions [39,40]. The proliferation of resource-intensive industries such as ferrous metal smelting, chemical production, thermoelectricity, and glass manufacturing escalates urban carbon emissions [6,10,41]. Capital-intensive industries that employ low-carbon production can aid in reducing carbon emissions. China's industrial carbon emission reduction strategies fall into three main categories. First is the development of budding industries such as microelectronics, software, and new energy, which can enhance city competitiveness and aid carbon reduction. Second, there is the expansion of low-carbon service industries like modern financial and information services, which are labor-intensive and deliver high-value addition. Third, there is the elimination of outdated production capacity, a vital step in early city development. After a certain point, cities shifting from resource-intensive industries to high-end industries becomes necessary as the scale effect of economic growth lessens [42]. Energy policy is another critical component of China's carbon emissions management as carbon dioxide emission from energy combustion forms the primary source [43]. Given that China's coal-dependent energy consumption structure

is not likely to change in the short term, enhancing efficiency emerges as the most expedient and effective solution to curtail fossil energy consumption.

Management-based policy policies encompass the regulation, auditing, and monitoring of urban carbon emission levels, exemplified by carbon audits, carbon monitoring, and early warnings, as well as carbon accounting [12]. Despite this, individuals managing policies may struggle to effectively control carbon emissions under prevailing conditions, primarily due to the significant implementation costs at the corporate level. Even so, management-based policy remains indispensable within the existing policy cluster system, bolstering and ensuring the successful execution of other types of policies [44]. Social policies involve promoting information disclosure, encouraging the operations of environmental social organizations, and launching low-carbon initiatives. Parallel to management-based policies, social policies lack the backing of high-level laws/regulations and mainly hinge on government-led demonstration actions, publicity drives, and ideological indoctrination, which might reduce their effectiveness.

In recent years, China has been actively promoting two directions of policy: economic policies and technology-oriented policies. Economic policies leverage market forces to alleviate the burden of policy implementation costs. These policies also offer carbon emitters more flexibility in designing emission reduction plans due to their certainty and predictability [44]. On the other hand, technology-oriented policies focus on low-carbon initiatives. They encompass scientific and technological advancements related to low-carbon practices, such as renovating buildings to be more energy-efficient, developing new energy sources, and implementing carbon sequestration and carbon capture technologies. Technology-oriented policies play a vital role in supporting other policy types and are crucial for achieving carbon emission reduction. They constitute the linchpin of carbon reduction efforts [25].

*2.3. Analytical Framework*

Achieving peak carbon requires the "decoupling" of economic and societal growth from carbon dioxide emissions. This implies that economic expansion should no longer correspond directly with increased carbon emissions. Extensive research has focused on the decoupling of carbon emissions from various developmental factors. Studies have considered the disconnection between industrial value added and carbon emissions in coastal cities [45], as well as the impact of carbon pricing on carbon emissions and economic growth [46]. The related research also includes decoupling carbon emissions from economic growth and decoupling carbon emissions from transportation [47].

The decoupling index offers valuable insight into the relationship between economic growth and carbon emissions in urban areas. Carbon emission reduction policies impact urban behaviors through numerous dimensions, forming a complex causal issue with multiple factors. These policy clusters should be systematically incorporated to analyze their combined effect on the decoupling of urban carbon emissions and economic growth.

Qualitative Comparative Analysis (QCA), a concept propounded by Ragin, is a method of analysis that lays emphasis on the case-oriented configuration, underpinned by Boolean algebra and set theory [48–50]. This method perceives the case and its contributing factors as an integrated unit, thereby allowing the discovery of the logical relationship between the configuration and outcomes of varying cases and factors through multi-factorial and multivariate case comparisons. It enables the identification of the structure of resultant variables and the formulation of various antecedent conditions that meet the desired outcomes [48,51]. The application of QCA to the study of policy combination issues in urban carbon emission decoupling is composed of two rationale: first, the complexity inherent in the decoupling of urban carbon emissions, given that the trajectory to achieve decoupling may originate from diverse policy combinations [52]; the second is QCA's ability to aptly address the asymmetry in causal relationships, which many quantitative analysis methods cannot, as they presuppose an assumption of causal effect consistency amongst the variables under research—denoting explanatory variable A (antecedent condition) as

the singular path to explained variable B (specific outcome). However, QCA postulates that the causal correlation during an event is non-consistent and non-linear, inferring that an assortment of factors and conditions could generate analogous results. These distinguishing characteristics of QCA facilitate a more profound comprehension of an event's complexity and diversity while uncovering the interplay amongst varying factors and their impacts under diverse conditions, hence providing policymakers with comprehensive, precise data. The challenge of urban carbon emission decoupling is immensely complex, and there is no single policy path to actualize the decoupling effects. Different policy mix combinations might result in urban carbon emissions decoupling [24]. Consequently, this study employs QCA to scrutinize the configurations and path choices fashioned by varying policy mixes in differing contexts.

This paper applies QCA to match the aforementioned six types of policies, explores which configuration of different types of policies promotes carbon emission decoupling, and constructs a carbon emission reduction policy mix analytical framework, as shown in Figure 2.

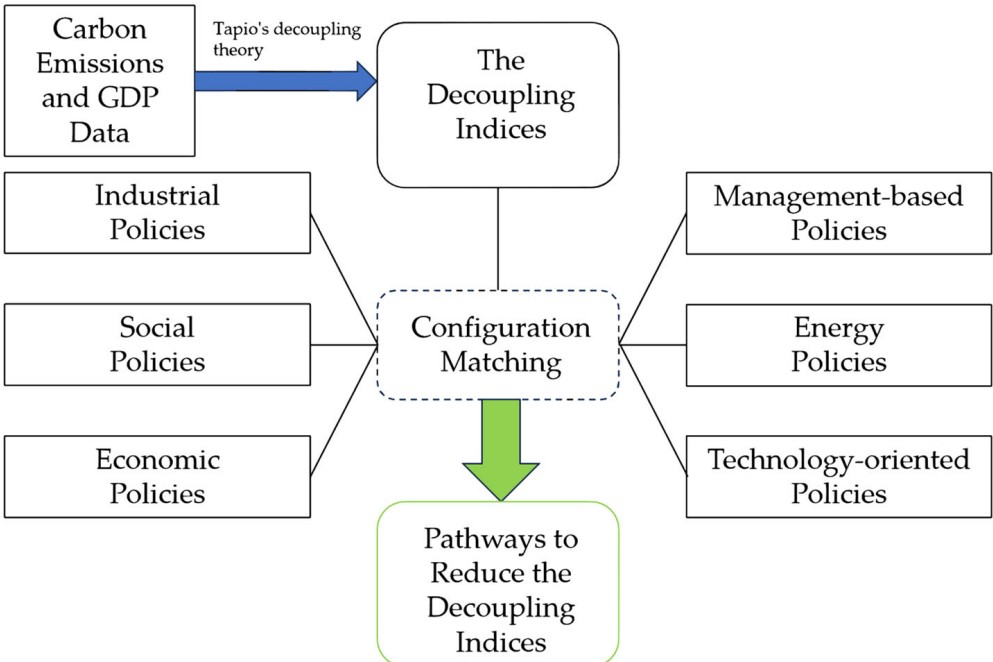

**Figure 2.** Research framework.

### 3. Data and Methods

#### 3.1. Measurement of the Decoupling Index

The frequently utilized decoupling indices include the OECD decoupling index and the Tapio elasticity coefficient method [53–55]. The OECD decoupling index, conceived in the World Economic Cooperation Organization's (WECO) 2002 report, presents some limitations tied to the choice of the base period that may skew the results. To address this, Tapio advanced the elasticity coefficient method, which introduces the concept of decoupling elasticity and elaborates on eight distinct decoupling states, which are illustrated in Figure 3 [54,55].

In the ideal state of strong decoupling, economic volume increases while total carbon emission volume diminishes, exhibiting the city's effective response to carbon emission reduction policies. In contrast, weak decoupling portrays both total carbon emission and economic volumes incrementing, but the former lags behind the latter. Recessive decoupling indicates a reduction in carbon emission volume at a faster rate than economic decline, suggesting that rigid economic policies may not be compatible with the city's economic direction. Negative decoupling indicates inadequate carbon emission reduction

strategies. Specifically, expansive negative decoupling indicates rapid growth in both carbon emissions and economic expansion, with carbon emissions outpacing economic development. Such a scenario exposes the ineffective carbon emission reduction policies and economic prioritization over the urban environment. Strong negative decoupling, the most detrimental urban development state, occurs when rising carbon emissions coincide with economic stagnation, leading to divergence. Unlike strong decoupling, weak negative decoupling indicates a slightly greater economic decline relative to carbon emissions. The expansive coupling state corresponds to simultaneous increases in carbon emissions and the city's economic aggregate, while recessive coupling state demonstrates contraction in both carbon emissions and economic growth [14].

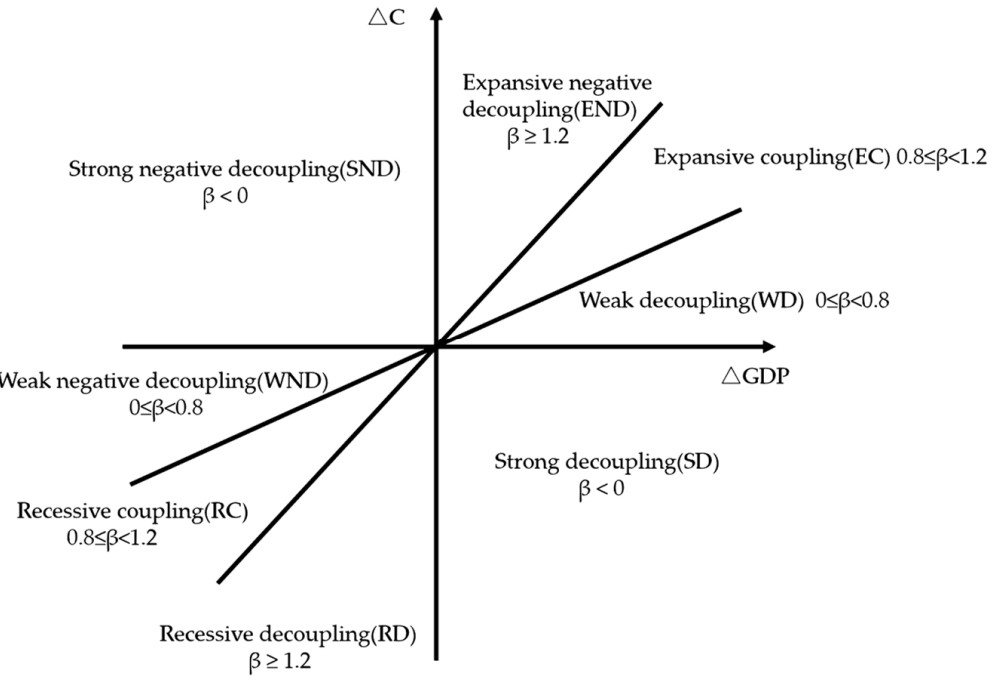

**Figure 3.** Tapio decoupling index system.

The Tapio decoupling elasticity provides an effective solution to overcome the issues associated with the base period choice encountered in the OECD indices [12]. For this reason, this study will utilize the Tapio decoupling elasticity method to quantitatively assess the magnitude of decoupling between the total carbon emissions and GDP across 41 cities in the YRD region [54,55]. The formula used for this calculation is as follows:

$$\beta = \frac{\Delta E / E}{\Delta Y / Y} \tag{1}$$

where $\beta$ is the decoupling elasticity coefficient, and $E$ and $Y$ denote the increase in carbon emissions and GDP, respectively. $\Delta E$ and $\Delta Y$ represent the carbon emission increase and GDP increase in two adjacent periods, respectively.

### 3.2. Qualitative Comparative Analysis

As QCA research broadens and deepens, numerous methods, such as clear set (csQCA), multivariate set (mvQCA), and fuzzy set (fsQCA), have evolved [56]. Often, fsQCA, in contrast to csQCA and mvQCA, is exceptional due to its ability to quantify the variable's effects on the outcome using membership score calibration. This quantitative technique allows researchers to comprehend more precisely the impacts these diverse conditional variables have on the outcome. Moreover, a consistency assessment crucially verifies the model's robustness, ensuring that the findings hold consistent across different datasets

and methodological choices. Subsequently, fsQCA can offer more dependable and credible research conclusions, which, in turn, prove instrumental in providing decisional support and references to policy-makers [57].

The operational steps to determine causal relationships using fsQCA entail [51,58], first, variable calibration. Grounded in set theory, it is essential to recalibrate each variable condition (for example, the type of carbon reduction policy) prior to applying QCA for computations. This involves assigning set memberships to variables, consequentially transforming them into fuzzy membership scores ranging from 0 to 1. Thus, points such as full-set membership, the intermediate-set (or crossover point) membership, and full-set non-membership must be set when recalibrating for set membership. This paper, taking cues from previous literature, calibrates six conditional variables and one result variable using the original data's 25%, 50%, and 75% values as calibration parameters [59].

The analysis's third step includes single-factor necessity analysis. Establishing causal conditions mandates identifying conditions necessary for the occurrence of a result. Necessity analysis tests whether certain carbon reduction policies are essential for urban carbon emissions to decouple. Hence, assessing individual conditions for necessity is required prior to configuration analysis. In QCA, the relationships of necessity and sufficiency between single conditional variables and outcome variables are investigated with the aid of consistency and coverage factors. Single variable necessity analysis's validity is determined using a consistency index, whose formula is simplified as:

$$Consistency(X_i \leq Y_i) = \sum[min(X_i, Y_i)] / \sum X_i \qquad (2)$$

The consistency value ranges between 0 and 1. However, in the real world, the outcomes of most events do not necessarily have a necessary condition. Therefore, based on existing research, this paper assumes that if $X$ (a single conditional variable or a combination of variables) is a necessary condition for $Y$, then the consistency value is greater than 0.9, and if condition $X$ is a sufficient condition for $Y$, then the consistency indicator is greater than 0.8. The coverage rate value is directly proportional to the degree to which $X$ explains $Y$. There is no specific threshold for its value, which may be relatively low in certain situations [50]. Its formula simplifies to:

$$Coverage(X_i \leq Y_i) = \sum[min(X_i, Y_i)] / \sum Y_i \qquad (3)$$

The fourth step in the QCA analysis is to carry out a standard analysis based on the truth table constructed previously. Through counterfactual analysis, we obtain three types of solutions: parsimonious, intermediate, and complex. If both parsimonious and complex solutions are used for counterfactual analysis at the same time, a parsimonious solution can be simplified, whereas using a parsimonious solution for counterfactual analysis simplifies to obtain an intermediate solution. If counterfactual analysis is not used at all, a complex solution will be obtained. Given that the complexity of the intermediate solution lies between that of the complex and parsimonious solutions, its conclusion takes into account both the representativeness of the complex solution and the universality of the parsimonious solution; hence, this paper mainly uses the results of the intermediate solution as the core explanatory solution.

### 3.3. Data Sources

3.3.1. Data Collection

This study utilizes data from 41 cities in the YRD region. The study sample is drawn from data accumulated between 2015 and 2020, derived from various sources, including the Anhui Statistical Yearbook, the Zhejiang Provincial Statistical Yearbook, the Jiangsu Provincial Statistical Yearbook, the Shanghai Municipal Statistical Yearbook, the China City Statistical Yearbook, and those of the aforementioned cities.

The effectiveness of carbon emission reduction policies is not unilateral; it necessitates interplay between policies spanning multiple fields to stimulate a policy synergy. Therefore,

policy documents focusing on carbon emissions reductions should encompass all policies relative to carbon emissions reduction across diverse sectors. The policies concerning carbon emission reduction in the YRD region are gathered from the magic weapon of the Peking University policy database and various government department portals, using keywords such as "carbon dioxide", "greenhouse gas", "carbon emissions reduction", and "carbon emissions control". From this search, a total of 389 policy documents were identified and compiled. In line with the research needs, this pool underwent a manual screening based on several criteria: the content of the policy document includes carbon management or carbon emissions reduction-related material, such as goals and plans; the document must include a clear policy subject and corresponding measures; and the policy document type aligns with the needs of this research. The screening process culminated in the selection of 252 carbon emissions reduction policy documents of high research value and effectively maintained relevance.

### 3.3.2. Selection of Variables

Cities within the YRD region are experiencing rapid economic development. Forecasts do not suggest an economic downturn soon. As such, the decoupling index is restricted to four of all eight states. In each case, lower values of the decoupling index for the outcome variable are more desirable. Thus, the more the decoupling index of the outcome variable decreases, the higher the value assigned, and conversely, the less it decreases, the lower the assigned value. Calibration anchor points are elucidated in Table 1.

**Table 1.** Calibration anchor points.

| Variant | Calibration Anchor | | |
| --- | --- | --- | --- |
| | **Full-Set Membership (75 Percent)** | **Intermediate-Set Membership (50 Percent)** | **Full-Set Non-Membership (25 Percent)** |
| Industrial policies (IP) | 33.415 | 22.896 | 12.933 |
| Energy policies (NP) | 33.437 | 18.222 | 9.674 |
| Management-based policies (MP) | 12.836 | 5.531 | 2.008 |
| Social policies (OP) | 8.229 | 6.256 | 3.568 |
| Economic policies (EP) | 23.155 | 11.021 | 5.887 |
| Technology-oriented policies (TP) | 40.547 | 25.310 | 15.082 |
| Change in decoupling index (TA) | 0.603 | 0.303 | 0.144 |

Source: Developed by the author using fsQCA3.0 software.

The unit of analysis in QCA is the combination of conditional variables. In this case, there are six conditional variables; therefore, logically, there would be $2^6$ (64) potential combinations of conditions. We need to construct a truth table to simplify the conditional variables by setting frequency thresholds and consistency thresholds. The purpose of setting the frequency threshold is to retain usable samples; in general, the larger the sample size, the higher the set value of the frequency threshold. Ragin suggests that when the consistency of the conditional variable with the outcome variable is not less than 75%, the conditional variable is the sufficiency configuration for the outcome variable. Therefore, this paper sets the threshold at 75% [51]. The consistency threshold shows the subset degree of the truth table row. This paper, referring to the research of Gong and Du, sets the consistency threshold of the number of cases at 0.75. The outcome variable that has not attained the configuration result is assigned a value of 1, and the others are assigned 0 [60]. The truth table is shown in Table 2.

**Table 2.** The truth table for policies of cities in the Yangtze River Delta (YRD) region.

| Serial Number | IP | NP | MP | OP | EP | TP | TA | N | PRI Consistency |
|---|---|---|---|---|---|---|---|---|---|
| 1 | 1 | 1 | 1 | 0 | 1 | 1 | 1 | 1 | 0.953 |
| 2 | 0 | 1 | 1 | 0 | 1 | 0 | 1 | 1 | 0.890 |
| 3 | 1 | 0 | 0 | 1 | 0 | 1 | 1 | 1 | 0.886 |
| 4 | 1 | 1 | 1 | 0 | 1 | 0 | 1 | 2 | 0.803 |
| 5 | 0 | 0 | 1 | 0 | 0 | 0 | 1 | 1 | 0.692 |
| 6 | 1 | 1 | 0 | 1 | 0 | 1 | 0 | 3 | 0.517 |
| 7 | 1 | 1 | 1 | 0 | 1 | 1 | 0 | 5 | 0.588 |
| 8 | 0 | 0 | 1 | 0 | 1 | 0 | 0 | 4 | 0.601 |
| 9 | 1 | 0 | 0 | 1 | 0 | 1 | 0 | 3 | 0.468 |
| 10 | 0 | 0 | 0 | 0 | 0 | 0 | 0 | 4 | 0.541 |
| 11 | 1 | 1 | 0 | 0 | 1 | 1 | 0 | 1 | 0.430 |
| 12 | 1 | 1 | 1 | 1 | 1 | 1 | 0 | 7 | 0.393 |
| 13 | 0 | 0 | 0 | 1 | 0 | 1 | 0 | 1 | 0.273 |
| 14 | 0 | 0 | 0 | 1 | 0 | 0 | 0 | 7 | 0.261 |

Source: Developed by the author using fsQCA3.0 software.

## 4. Results

### 4.1. Decoupling Indices

The decoupling states of each city in the YRD region calculated using Formula (1) are shown in Figure 4. Overall, the relationship between carbon emissions and development is relatively positive. Most cities exhibit a weak decoupling state ($0 \leq \beta < 0.8$), with Hangzhou, Huzhou, and Shanghai showing more ideal decoupling indices. This finding aligns with the research results of Li Z [61] and Li Y [62]. There are also some differences in the results for certain cities due to the use of different indicators such as population and urbanization.

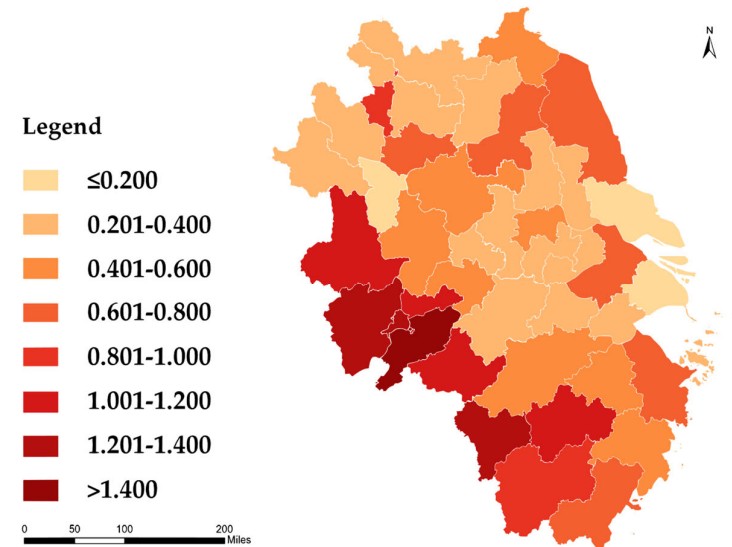

**Figure 4.** The decoupling indices and states of cities in YRD region.

### 4.2. Necessity Analysis of Single Conditional Variables

In condition necessity, the results hinge on the consequential existence of specific conditional variables. The absence of these conditions effectively guarantees that the intended result will not manifest. Examining these necessary conditions individually can discern if a single antecedent condition can yield a decrease in the city's carbon emission decoupling index. Consequently, prior to employing a fuzzy set truth table for configuration analysis, such an examination proves invaluable. The software fsQCA 3.0 was employed to evaluate the necessity of each conditional variable. The results, computed

using Formulas (2) and (3), are summarized in Table 3. Observably, the homogeneity of the six antecedent conditions impacting the carbon emission decoupling in 41 YRD cities barely reaches 0.9, in line with the methodology assumptions. This signals a poor correlation between the initiation of any of the six carbon reduction policies and the achievement of urban carbon emission decoupling. In essence, urban carbon emission decoupling is a direct consequence of an amalgamation of multiple policies, rather than a single policy. Moreover, none of these carbon reduction policies exhibit a consistency exceeding 0.8, again adhering to the methodology assumptions. This suggests that declaring any of these policy types as either necessary or sufficient conditions for achieving intended results would be premature. A more meticulous exploration of how different policy combinations influence urban carbon emissions decoupling is therefore recommended.

**Table 3.** Results of univariate necessity analysis.

| Conditional Variable | Consistency | Coverage |
|:---:|:---:|:---:|
| IP | 0.762880 | 0.543787 |
| ~IP | 0.613278 | 0.517166 |
| NP | 0.590698 | 0.605202 |
| ~NP | 0.519984 | 0.494488 |
| MP | 0.596019 | 0.576874 |
| ~MP | 0.468271 | 0.470905 |
| OP | 0.436334 | 0.449381 |
| ~OP | 0.662574 | 0.627063 |
| EP | 0.598986 | 0.583938 |
| ~EP | 0.480140 | 0.479265 |
| TP | 0.725810 | 0.528386 |
| ~TP | 0.558261 | 0.540704 |

Note: ~ represents negation. Source: Developed by the author using fsQCA3.0 software.

### 4.3. Sufficiency Analysis of Antecedent Configurations

Through truth table analysis, three types of results can be obtained: parsimonious solutions, intermediate solutions, and complex solutions. This paper, based on parsimonious and intermediate solutions, uses fuzzy set configuration diagrams to judge and distinguish the core and peripheral conditions that lead to the decoupling of urban carbon emissions, as presented in Table 4.

**Table 4.** Configuration analysis of carbon emission reduction policies in cities in the YRD region.

| Conditional Variable | Configuration 1 | Configuration 2 | Configuration 3 | Configuration 4 |
|:---:|:---:|:---:|:---:|:---:|
| Industrial policies (IP) | ● | ⊗ | ● | ● |
| Energy policies (NP) | ● | ● | ⊗ | ● |
| Management-based policies (MP) | ● | ● | ⊗ | ● |
| Social policies (OP) | ⊗ | | ● | ⊗ |
| Economic policies (EP) | | ⊗ | ⊗ | ● |
| Technology-oriented policies (TP) | ⊗ | ● | ● | ● |
| Consistency | 0.868616 | 0.830080 | 0.851692 | 0.895942 |
| Raw coverage | 0.198413 | 0.183606 | 0.158691 | 0.142579 |
| Unique coverage | 0.069287 | 0.034257 | 0.064729 | 0.011424 |
| Overall consistency (solution consistency) | | 0.837831 | | |
| Overall coverage (solution coverage) | | 0.25976 | | |

Note: ● indicates core condition present; • indicates peripheral condition present; ⊗ indicates core condition absent; ⊗ indicates peripheral condition absent; a space indicates condition possibly either present or absent. Source: Developed by the author using fsQCA3.0 software.

The presented table offers a cogent representation of the relative significance of each antecedent condition for configurations, distinguishing core from peripheral conditions. It uses a black solid circle to signify the existence of a condition within a configuration, a

circle intersected by "x" if the condition is absent, and a blank space to denote irrelevance of the respective condition. The larger circle refers to the shared presence of a condition in both intermediate and parsimonious solutions, whilst a smaller circle represents peripheral conditions necessary for the realization of the intermediate configuration. Four configurations contributing to urban carbon emission decoupling emerge, boasting consistencies of 0.868616, 0.830080, 0.851692, and 0.895942, surpassing the widely accepted theoretical threshold of 0.8. This implies that these configurations may serve as sufficient conditions for explaining policy combinations that facilitate urban carbon emission decoupling.

## 5. Discussion

The core conditions of configuration 1 incorporate the emergence of energy policies and an absence of technology-oriented policies. In contrast, the presence of industrial and management-based policies, and the absence of social ones, constitute peripheral conditions. The consistency of configuration 1 is 0.868616, and the raw coverage rate is 0.198413, indicating that this configuration can explain 19.8% of the cases. This configuration uses Shanghai as a representative case. Industrial and management-based policies mainly function as peripheral conditions in Configuration 1, reflecting the research by Gong et al. [24]. Here, it is evident that energy policies significantly contribute to the reduction of Shanghai's carbon emission decoupling index. Carbon emissions are directly affected by fossil fuel consumption. Despite Shanghai's limitations in non-fossil energy development, efforts to optimize the energy structure continue. The auxiliary antecedent variables in this configuration consist of industrial and management-based policies. The former represents the improvement of the city's industrial structure, while the latter indicates the rigor in city management inspection and performance appraisal. Shanghai, in recent years, has been transitioning steadily towards an industry upgrade helmed by service and high-tech sectors, thereby reducing its reliance on fossil fuels. The implementation of stringent inspection and performance appraisals has bolstered the execution of energy policies.

The presence of the technology-oriented policy and the absence of the economic policy serve as the key conditions of configuration 2, while the remaining conditions are supplementary. The consistency of configuration 2 is 0.830080, and the raw coverage rate is 0.183606, indicating that this configuration can explain 18.3% of the sample cases. Therefore, it is inferred that this configuration, led by the technology-oriented policy and supplemented by management-based and energy policies, can facilitate cities in achieving economic decoupling from carbon emissions, even when the enforcement of economic and industrial policies is lax. As such, Wuxi is an exemplary case. Based on policy text assessments, Wuxi has an industry that meets international leading standards, leaving little room for carbon reduction within the production processes. However, this suggests the possibility for cities to counteract the effects of industrial development's energy consumption through augmenting low-carbon technology levels and optimizing energy structures. This, in turn, would expedite urban carbon emission decoupling.

Core conditions of configuration 3 are defined by the existence of industrial policy and the absence of management-based policy. The consistency of configuration 3 is 0.851692, and the raw coverage rate is 0.158691, indicating that this configuration can explain 15.8% of the sample cases. Peripheral conditions are constituted by the emergence of technology-oriented policy, the deficiency of energy policy and economic policy, and the rise of social policy. Jiaxing, reflecting these findings, rigorously initiated the development of a digital economy system. This configuration embodies a city primarily molded by industrial policies and supplemented by technology-oriented policies. Such a city can circumvent its reliance on management to an extent by investing in high-tech industries with low input and high output, consequently generating reduced pollution emissions. This obviates the need for excessive regulatory investment, facilitating a decrease in the city's carbon emission decoupling index.

The introduction of industrial and energy policies represents the crux of the configuration 4. The non-emergence of management-based policies along with the introduction

of economic and technology-oriented policies comprise the supportive conditions. The consistency of configuration 4 is 0.895942, the highest among the four configurations, indicating that the likelihood of a decline in a city's carbon emission reduction decoupling index is highest when the policy combination under this configuration is formed. The raw coverage rate is 0.142579, indicating that this configuration can explain 14.2% of the city's carbon emission reduction decoupling phenomenon. Configuration 4 is exemplified by the city of Ningbo. The primary strength of this configuration lies in the robust utilization of varied strategies to exploit urban carbon reduction potential. Ningbo's implementation of numerous industrial and energy carbon reduction policies during the research period illustrates a distinct path-dependent character in carbon reduction propagation. Industrial structure optimization and energy saving measures provide clear strategies to improve carbon emission efficiency, maintaining a promising avenue to reduce citywide carbon emissions. The research findings from Guo validate the necessity of carbon monitoring systems, corroborating the results for configurations 1, 2, and 4 from this study [11]. By incorporating a comparative study between central western and eastern China conducted by Gong et al. [24], it is clear that regional differences contribute to the variances in study outcomes. Social policies, which are absent in three configurations and auxiliary in one, play a more nominal role, aligning with the findings of Brouhle et al. [63].

## 6. Conclusions

Through examining the aforementioned configurations, it becomes evident that industrial and technology-oriented policies currently dominate as the primary influences guiding cities towards carbon emission decoupling. Presently, as cities develop, optimizing industrial structures and upgrading technology becomes key in enhancing economic efficiency, which sufficiently counters the effects of growing carbon emissions. The four configurations hold a minimum of three and a maximum of five policy types. This transformation is fostered by the intensified focus on urban conditions and the persistent enhancement of government regulation. Within the four structures, management-based policies only serve as supporting components. Social and economic policies have yet to take a leading position in managing carbon emissions, adopting supporting roles in a singular configuration each. For instance, social policies are notably absent from three configurations, suggesting their current limited effect on advancing urban carbon reduction initiatives. Additionally, the influence of economic policies on the decoupling of urban carbon emissions remains suboptimal.

Based on the outcomes identified above, this study proposes two overarching recommendations for policy formulation. First, enhance the carbon emission reduction framework by incorporating "dual carbon" objectives into the legislations of corresponding departments. This involves implementing low-carbon projects within existing environmental resources and energy laws, thereby building a legal structure that addresses various facets of carbon reduction. Simultaneously, it is crucial to devise laws and regulations that support economic policies such as carbon taxation and trading. Locally implemented supervision and punitive measures are essential for ensuring adherence to carbon trading quotas. This system should be progressively refined and expanded to improve the city's overall carbon trading outcomes. Furthermore, we recommend improving the economic policy support systems like green finance and credit, speeding up the development of carbon financial markets, and reducing the costs associated with low-carbon transformations for businesses. Second, based on each city's level of development, tailored carbon emission reduction policies must be formulated. For less developed cities, the focus is on reinforcing the creation of industry-specific and energy policies, optimizing second-tier industry growth, eliminating obsolete capacity, tightening standards for industrial transfer, and fostering simultaneous resolution of production capacity and industrial upgrades. These actions will perpetually promote low-carbon transformation in key emission industries and the establishment of low-carbon industrial parks, thereby preventing any undue increase in the rate of carbon emission growth vis-a-vis economic growth. For economically advanced cities,

efforts should be geared towards hastening green, low-carbon technological innovation, replacing traditional technologies with low-carbon alternatives, increasing R&D investment in universities and pertinent research bodies, bolstering policy incentives and guidance, and hastening breakthroughs in zero-carbon energy technology and carbon emission end-processing technology. In tandem, the need for consolidating industry–academia–research partnerships and promoting the application of low-carbon technology cannot be overstated.

This research contributes in two ways. Firstly, it addresses a gap in the current literature which recognizes the importance of policy combinations in carbon emission reduction but largely focuses on the collective impacts of such policies, overlooking the value of individually analyzing diverse policy configurations. To tackle this issue, our study proposes a framework to scrutinize the effects of various carbon reduction policy combinations using the Tapio decoupling index–fsQCA configuration matching method. This framework elucidates the areas where specific policy categories have had substantial impacts and areas which offer significant prospects for enhancement, thus offering insights for future research on carbon reduction policy combinations. Secondly, by employing the Yangtze River Delta region as a case study and applying our proposed research framework, we provide practical and evidence-based references for future policy development and optimization in analogous regions.

This study acknowledges certain limitations: At the outset, we focus on a wider, macro scale to evaluate the influence of various policies combinations on implementation effectiveness within an expansive geographical area, encompassing a complex range of sectors like energy, economics, and environmental ecology. This research is grounded on publicly accessible policy documents and statistical data. Constraints of time, resources, and the wider scope of research preclude us from detailed micro-analysis of the specific impacts of these policies. Furthermore, our analytical framework prioritizes policies over policy tools. Although a single policy may comprise several policy tools, our study did not utilize these as an analytical medium, potentially influencing the granularity of our results. We observe potential for further empirical research to bridge these gaps. For instance, one could investigate the mechanisms of carbon emission impacts with relation to the implementation of combined policies from the viewpoint of policy tools for specific industries.

**Author Contributions:** Conceptualization, C.B.; methodology: P.X. and Q.X.; formal analysis: Q.X.; writing: Q.X.; funding acquisition: C.B. All authors have read and agreed to the published version of the manuscript.

**Funding:** This research was supported by the Shanghai Planning Office of Philosophy and Social Science (No. 2021XSL024).

**Institutional Review Board Statement:** Not applicable.

**Informed Consent Statement:** Not applicable.

**Data Availability Statement:** The datasets used and/or analyzed during the current study are available from the corresponding author on reasonable request.

**Conflicts of Interest:** The authors declare no conflict of interest.

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
