# Peer review of "Multifaceted Pathways of Carbon Emission Reduction Policies: A Study Based on the Decoupling Effect and fsQCA Method in the Yangtze River Delta Urban Agglomeration"

_sustainability, doi:10.3390/su151813745_

Round 1

Reviewer 1 Report

Multifaceted Pathways of Carbon Emission Reduction Policies was studied in this paper. The following issues must be addressed before publication:
1. Abstract: Give some quantitative conclusions about the description of the results.
2. The introduction seems to be light and not rich enough. Some key papers about the carbon emission should be discussed:
1) Characteristics, mechanisms, and environmental LCA of WMA containing sasobit: An analysis perspective combing viscosity-temperature regression and interface bonding strength, https://doi.org/ 10.1016/j.jclepro.2023.136255
3. The text of the body should be aligned at both ends.
4. Equations should be mentioned in the main text and given the references.
5. The clarity of Figures should be improved, and vector graphics are recommended.
6. Moderate editing of English language is required.
7. In discussion: Barely see the figures, lack of immediacy.
8. Limitations of the study should be appropriately mentioned in the conclusion

Moderate editing of English language required.

Reviewer 2 Report

Please read the comment.

Please read the comment.

Reviewer 3 Report

The paper is written academically and makes a contribution to sustainable development. The largest emitters of carbon gases into the atmosphere are highly developed economies. Therefore, it is the task of these economies to look at various possibilities for reducing carbon dioxide emissions, which is certainly what China, as a strong economic power, is doing. The authors provided research that can really serve to find adequate economic, technological, energy and management strategies, when reducing CO2 emissions, depending on the region in which the cities are located and in accordance with the respective needs of the cities. 
